# Metabolic Effects of Sodium Thiosulfate During Resuscitation from Trauma and Hemorrhage in Cigarette-Smoke-Exposed Cystathionine-γ-Lyase Knockout Mice

**DOI:** 10.3390/biomedicines12112581

**Published:** 2024-11-12

**Authors:** Maximilian Feth, Felix Hezel, Michael Gröger, Melanie Hogg, Fabian Zink, Sandra Kress, Andrea Hoffmann, Enrico Calzia, Ulrich Wachter, Peter Radermacher, Tamara Merz

**Affiliations:** 1Department of Anesthesiology, Critical Care, Emergency Medicine and Pain Therapy, German Armed Forces Hospital Ulm, 89081 Ulm, Germany; 2Institute for Anesthesiological Pathophysiology and Process Engineering, Ulm University, 89081 Ulm, Germanyulrich.wachter@uni-ulm.de (U.W.); peter.radermacher@uni-ulm.de (P.R.); 3Department of Anesthesiology and Intensive Care Medicine, University Hospital Ulm, 89081 Ulm, Germany

**Keywords:** hydrogen sulfide, kidney function, glucose oxidation, gluconeogenesis, ureagenesis, proteolysis, oxidative phosphorylation, electron transfer capacity

## Abstract

Background: Acute and chronic pre-traumatic cigarette smoke exposure increases morbidity and mortality after trauma and hemorrhage. In mice with a genetic deletion of the H_2_S-producing enzyme cystathione-γ-lyase (CSE^−/−^), providing exogenous H_2_S using sodium thiosulfate (Na_2_S_2_O_3_) improved organ function after chest trauma and hemorrhagic shock. Therefore, we evaluated the effect of Na_2_S_2_O_3_ during resuscitation from blunt chest trauma and hemorrhagic shock on CSE^−/−^ mice with pre-traumatic cigarette smoke (CS) exposure. Since H_2_S is well established as being able to modify energy metabolism, a specific focus was placed on whole-body metabolic pathways and mitochondrial respiratory activity. Methods: Following CS exposure, the CSE^−/−^ mice underwent anesthesia, surgical instrumentation, blunt chest trauma, hemorrhagic shock for over 1 h (target mean arterial pressure (MAP) ≈ 35 ± 5 mmHg), and resuscitation for up to 8 h comprising lung-protective mechanical ventilation, the re-transfusion of shed blood, fluid resuscitation, and continuous i.v. noradrenaline (NoA) to maintain an MAP ≥ 55 mmHg. At the start of the resuscitation, the mice randomly received either i.v. Na_2_S_2_O_3_ (0.45 mg/g_bodyweight_; n = 14) or the vehicle (NaCl 0.9%; n = 11). In addition to the hemodynamics, lung mechanics, gas exchange, acid–base status, and organ function, we quantified the parameters of carbohydrate, lipid, and protein metabolism using a primed continuous infusion of stable, non-radioactive, isotope-labeled substrates (gas chromatography/mass spectrometry) and the post-mortem tissue mitochondrial respiratory activity (“high-resolution respirometry”). Results: While the hemodynamics and NoA infusion rates did not differ, Na_2_S_2_O_3_ was associated with a trend towards lower static lung compliance (*p* = 0.071) and arterial PO_2_ (*p* = 0.089) at the end of the experiment. The direct, aerobic glucose oxidation rate was higher (*p* = 0.041) in the Na_2_S_2_O_3_-treated mice, which resulted in lower glycemia levels (*p* = 0.050) and a higher whole-body CO_2_ production rate (*p* = 0.065). The mitochondrial respiration in the heart, kidney, and liver tissue did not differ. While the kidney function was comparable, the Na_2_S_2_O_3_-treated mice showed a trend towards a shorter survival time (*p* = 0.068). Conclusions: During resuscitation from blunt chest trauma and hemorrhagic shock in CSE^−/−^ mice with pre-traumatic CS exposure, Na_2_S_2_O_3_ was associated with increased direct, aerobic glucose oxidation, suggesting a switch in energy metabolism towards preferential carbohydrate utilization. Nevertheless, treatment with Na_2_S_2_O_3_ coincided with a trend towards worsened lung mechanics and gas exchange, and, ultimately, shorter survival.

## 1. Background

Hydrogen sulfide (H_2_S), which is referred to as the “*third gaseous mediator*” [1], has various biological functions [1,2]. Moreover, depending on its local concentration, it may either inhibit or stimulate mitochondrial respiratory activity. Therefore, the role of H_2_S has been investigated in various models of circulatory shock. In rodent models of hemorrhagic shock, both the endogenous formation and the exogenous administration of H_2_S have yielded equivocal results, whether beneficial, detrimental, or no effects at all [3,4,5,6,7,8]. Issues regarding therapeutic H_2_S administration comprise the pulmonary toxicity of inhaled H_2_S; the generation of unphysiologically high, and thus potentially toxic, peak concentrations of H_2_S by sulfide salts; and undesired hemodynamic side effects of the so-called “*slow-releasing H_2_S donors*” [7,9,10]. In contrast, sodium thiosulfate (Na_2_S_2_O_3_) is an exogenous H_2_S donor that is already clinically established and devoid of major side effects [11,12,13,14]. Previously, Na_2_S_2_O_3_ was shown to mitigate acute lung injury, thereby improving the survival of mice after an LPS injection with cecal ligation and puncture-induced polymicrobial sepsis [15,16]. In line with these findings, we recently observed that Na_2_S_2_O_3_ prevented the otherwise progressive deterioration of lung mechanics and pulmonary gas exchange induced by hemorrhagic shock and resuscitation in swine with pre-existing coronary artery disease, a condition associated with the impaired expression of the key cardiovascular H_2_S-generating enzyme cystathione-γ-lyase (CSE) [17,18]. We confirmed these beneficial effects of Na_2_S_2_O_3_ in a resuscitated model of combined blunt chest trauma and hemorrhagic shock in mice with a genetic CSE deletion (CSE knockout, CSE^−/−^) [19]. In that study, besides the attenuation of acute lung and kidney injuries, as well as improvements in hemodynamic stability, as demonstrated by the reduced norepinephrine requirements to maintain the mean arterial pressure, a Na_2_S_2_O_3_ treatment was associated with lower endogenous glucose production, suggesting a reduction in both hepatic and renal metabolic demand. However, in contrast to these observations, the Na_2_S_2_O_3_ treatment did not exert any beneficial effect in the same murine model when CSE^−/−^ mice with streptozotocin-injection-induced diabetes type 1 were investigated. This highlights the translational importance of investigating underlying co-morbidities in shock research, not only in clinical scenarios, but also in animal models [20].

It is well established that pre-traumatic cigarette smoke (CS) exposure aggravates post-traumatic acute lung injuries and that both CSE and H_2_S protect against CS-exposure-induced chronic obstructive lung disease (COPD) [21,22]. Moreover, in our murine model of blunt chest trauma, combining a CSE genetic deletion and pre-traumatic CS exposure not only aggravated a post-traumatic lung injury, but was also associated with lower glycemia levels than either the CSE deletion or CS exposure alone [23]. Interestingly, exogenous H_2_S administration using the “*slow-releasing H_2_S donor*” GYY4137 restored blood glucose levels to normal in the CSE^−/−^ mice [24]. The administration of Na_2_S_2_O_3_ as a H_2_S donor also mitigated LPS-induced liver injury in wild-type animals, and the attenuation of acute liver failure in CSE^−/−^ mice coincided with increased thiosulfate levels [25]. Finally, and most recently, Na_2_S_2_O_3_ was shown to prevent tissue damage in liver ischemia–reperfusion injury by increasing the organ antioxidant pool, thereby protecting mitochondrial function. Hence, many studies have demonstrated that Na_2_S_2_O_3_ potentially exerts its organ-protective effects via H_2_S-related metabolic modulation. Consequently, we hypothesized that Na_2_S_2_O_3_ might beneficially influence energy metabolism, and thereby possibly mitigate organ dysfunction during resuscitation from trauma and hemorrhage in CS-exposed CSE^−/−^ mice [26]. CSE^−/−^ mice with a pre-traumatic CS exposure were studied, because pre-traumatic CS exposure per se is associated with lowered blood H_2_S levels; in other words, the beneficial effects of Na_2_S_2_O_3_ administration on mice were more likely to be obtained [27]. Moreover, we assumed that these mice would be particularly susceptible to an enhanced metabolic demand; CS exposure with more pronounced systemic hyper-inflammation may cause both enhanced glycolytic activity and increased glucose oxidation, whereas right heart dysfunction due to an increased right ventricular afterload induced by lung hyperinflation may impair the liver’s metabolic capacity [23]. Irrespective of any genetic CSE deletion, the energy metabolism was assessed through the combined quantification of metabolic pathways using stable, i.e., non-radioactive, isotope substrate labeling and the determination of mitochondrial respiration using “high-resolution respirometry” [7,19,20,28]. To strengthen the translational value of the study, all the mice were treated with full-scale intensive care according to the current guidelines for the acute management of trauma and hemorrhage, and Na_2_S_2_O_3_ was administered using a “post-treatment” design, i.e., after the initiation of resuscitation.

## 2. Materials and Methods

### 2.1. Materials

A comprehensive list of the materials, devices, chemicals, and drugs used for this study is provided in the online Appendix A.

### 2.2. Study Animals

This study was approved by the University of Ulm Animal Care Committee and the Federal Authorities for Animal Research (Regierungspräsidium Tübingen Baden-Württemberg, Germany, Reg.-Nr. 1387, approved 31 January 2018). All the experiments were performed in accordance with the National Institutes of Health Guidelines on the Use of Laboratory Animals and the European Union’s “Directive 2010/63 EU on the protection of animals used for scientific purposes”. The present experiments were part of a larger protocol that also included mice without co-morbidities and streptozotocin-induced diabetes type 1 [19,20]. Since the federal authorities only approved the animal experiments for CSE^−/−^ mice, in which the strongest effect of Na_2_S_2_O_3_ was expected, we were not allowed to include wild-type control mice in the present study as originally planned. The animals were kept under standardized conditions, with free access to water and food.

Twenty-eight male homozygous (CSE^−/−^)-deficient mice (C57BL/6J.129SvEv) [29], bred in-house and aged 12–16 weeks with a body weight of 25–45 g, were randomly assigned to the control (“vehicle”) or sodium thiosulfate (“Na_2_S_2_O_3_^”^) groups.

### 2.3. Cigarette Smoke Inhalation Procedure

Cigarette (Roth-Händle without filters, tar: 10 mg, nicotine: 1.0 mg, carbon monoxide: 6 mg, Badische Tabakmanufaktur Roth-Händle^®^, Lahr, Germany) smoke (CS) exposure was performed using a standardized protocol as described previously. For a maximum of 5 days per week over a period of 3–4 weeks, in a heated exposure chamber, the animals were exposed for over 15 min per cigarette to 4 cigarettes on day 1, 6 on day 2, and 8 on the subsequent days. Each exposure was followed by 8 min of access to fresh air (15 L/min), and an additional 24 min break after every other cigarette [23,27,30,31]. For this purpose, a semi-automatic cigarette lighter and a smoke generator with an electronic timer were used to control the exposure (Boehringer Ingelheim Pharma GmbH & Co. KG, Biberach, Germany). The particle concentration was monitored using a real-time ambient particle monitor (MicroDust Pro, Casella, Amherst, NH, USA). The general animal health and weight were assessed daily during the exposure period, allowing for an individualized smoke exposure regimen and subsequently reducing animal harm. In previous studies, this CS-exposure procedure did not cause any effects on the behavior, body weight, or respiratory patterns of the mice; in addition, all the experimental groups were exposed to a comparable total CS dose. After the CS exposure, the mice were transported from Boehringer Ingelheim (Biberach, Germany) to Ulm University and were allowed to recover for 1 week after arrival to avoid acute stress effects induced by the CS procedure and/or the transport [27,30,31].

### 2.4. Anesthesia, Blunt Chest Trauma, Surgical Instrumentation, and Hemorrhagic Shock

The procedures for anesthesia, blast-wave-induced blunt chest trauma, surgical instrumentation, the induction of hemorrhagic shock, and subsequent resuscitation were performed as previously described [19,20,31]. After the induction of anesthesia (buprenorphine, 1,5 µg/g_bodyweight_ s.c.; inhaled sevoflurane, 2.5%), the animals underwent single-blast-wave-generated blunt chest trauma followed by placement on an instrumentation bench connected to a closed-loop system for body temperature control and maintenance at 37 °C. An i.p. injection of 120 µg/g_bodyweight_ of ketamine (Ketanest-S, Pfizer, New York City, NY, USA), 0.25 µg/g_bodyweight_ of fentanyl (Fentanyl-hameln, Hameln Pharma Plus GmbH, Hameln, Germany), and 1.25 µg/g_bodyweight_ of midazolam (Midazolam-ratiopharm, Ratiopharm, Ulm, Germany) was administered and surgical instrumentation was performed, comprising a tracheostomy with the subsequent initiation of “*lung-protective*” mechanical ventilation and the placement of catheters in the jugular vein, the carotid, the femoral artery, and the urinary bladder [19,20,28,31]. After the placement of the catheter in the jugular vein, anesthesia was maintained using a continuous infusion of 30 µg/h/g_bodyweight_ of ketamine and 0.3 µg/h/g_bodyweight_ of fentanyl. Lung-protective mechanical ventilation was performed using a pressure-controlled mode, with the ventilator settings being a fraction of inspired O_2_ (F_I_O_2_) at 0.21, a respiratory rate of 150/min, a tidal volume of 6 μL/g_bodyweight_, and an inspiratory/expiratory time ratio of 1:2. The respiratory rate was modified to maintain an arterial PCO_2_ (PaCO_2_) of 35–45 mm Hg, and the positive end-expiratory pressure (PEEP) was adjusted according to the arterial PO_2_ (PaO_2_/FiO_2_-ratio > 300 mm Hg: PEEP = 3 cmH_2_O; PaO_2_/FiO_2_-ratio < 300 mm Hg: PEEP = 5 cm H_2_O; PaO_2_/FiO_2_-ratio < 200 mm Hg: PEEP = 8 cm H_2_O) [23,27]. Recruitment maneuvers (5 s hold at 18 cm H_2_O) were repeated hourly to avoid the impairment of thoraco-pulmonary compliance due to anesthesia- and/or supine-position-induced atelectasis [32]. Subsequently, the animals underwent a 60 min period of hemorrhage through the removal of 30 µL/g_bodyweight_ of blood via the femoral arterial line. Thereafter, the mean arterial pressure (MAP) was titrated to 35 ± 5 mm Hg through a further blood withdrawal and/or the re-transfusion of a blood bolus (50 μL each). After one hour, resuscitation was started, comprising the re-transfusion of the shed blood, the administration of hydroxyethyl starch (Tetraspan^®^, Braun, Melsungen; 20 µL/h/g_bodyweight_) and, if necessary, norepinephrine to maintain an MAP > 55 mm Hg. Immediately upon the initiation of resuscitation, the mice randomly received either a bolus injection of Na_2_S_2_O_3_ (0.45 mg/g_bodyweight_) or the same amount of a vehicle solution (0.9% NaCl) [19,20]. At the end of the experiment, the mice were exsanguinated and blood and tissue samples were collected for further analysis. Figure 1 provides a general overview of the experimental setup, timeline, anesthesia, surgical preparation, and experimental protocol.

### 2.5. Parameters of Lung Mechanics, Hemodynamics, Gas Exchange, and Acid–Base Status

The core temperature, hemodynamics, and lung mechanics were recorded hourly (LabChart, ADInstruments, Colorado Springs, CO 80907, USA). Thoraco-pulmonary compliance was obtained automatically using the respirator (Flexivent small animal ventilator, Scireq, Montreal, QC, Canada). Arterial blood samples were taken at baseline (i.e., immediately after the insertion of the arterial catheter) and at the end of the experiment.

### 2.6. Metabolism and Organ Function

The arterial lactate and glucose levels were measured at baseline (i.e., immediately after the insertion of the arterial catheter) and at the end of the experiment. Together with the urine output, a gas chromatography/mass spectrometry (GC/MS) measurement of the plasma und urinary creatinine concentrations using ^2^H_3_-creatinine (CDN isotopes, Pointe-Claire, QC H9R1H1, Canada) as the internal standard allowed the level of creatinine clearance to be calculated according to the following formula:Creatinine clearance = Creatinine_Urine_ · Volume_Urine_/Creatinine_Plasma_ · t(1)
where Creatinine_Urine_, Creatinine_Plasma_, Volume_Urine_, and t are the urinary and plasma creatinine concentrations, the volume of sampled urine, and the urine sampling time, respectively [19,20,28].

Endogenous glucose production and direct, aerobic glucose oxidation, as well as the production rates of urea, glycerol, and leucine as markers of the hepatic metabolic capacity, lipolysis, and protein degradation, were determined during a primed continuous intravenous infusion of ^13^C_6_-glucose, ^15^N_2_-urea, ^2^H_5_-glycerol (Campro Scientific, Berlin, Germany), and 5,5,5-^2^H_3_-leucine (CIL, Tewksbury, MA, USA), as described in detail previously [19,20,28].

The direct, aerobic glucose oxidation rate was assessed from the expiratory ^13^CO_2_ release, determined from the total expiratory CO_2_ release and the ^13^CO_2_ enrichment. In addition to creatinine, the plasma levels of glucose and urea were determined. After thawing the samples, they were spiked with internal standards (6,6-^2^H_2_-glucose, ^2^H_3_-creatinine, and N-methyl-urea for the quantification of glucose, creatinine, and urea), deproteinized, and purified using cation-exchange solid-phase extraction (the separation of creatinine from interfering creatine and creatinephosphate). The next steps were derivatization and a GC/MS analysis. Glucose and glycerol were derivatized with N-methyl-bis(trifluoroacetamide) (MBTFA) to the corresponding trifluoroacetates. Urea and leucine were converted with N-(*tert*-butyldimethylsilyl)-N-methyltrifluoroacetamide (MTBSTFA) to *tert*-butyl-dimethylsilyl derivatives. Creatinine was analyzed as a trimethylsilyl derivative after a reaction with N,O-bis(trimethylsilyl)trifluoroacetamide (BSTFA). The GC/MS determinations were performed with an Agilent 6890/5973 GC/MS system housing an MN Optima-5-MS capillary column (12 m × 0.2 mm, 0.35 µm film thickness; Macherey-Nagel, Düren, Germany). The MS was operated using electron impact ionization in the selected ion-monitoring mode. The plasma concentrations of glucose, urea, and creatinine, as well as the urinary creatinine concentration, were determined with a six-point calibration curve. The peak area ratios of the endogenous compound vs. the added internal standard were plotted against the amount ratios to generate calibration curves. The rates of appearances (endogenous glucose, i.e., the gluconeogenesis, glycerol, urea, and leucine production rates, respectively) were calculated according to Equation (2) from the isotope infusion rates and the measured ratio of labeled vs. unlabeled compounds in the plasma:Ra = Inf/TTR(2)
where Ra is the rate of appearance, Inf is the infusion rate of the stable isotope-labeled compound, and TTR is the tracer (isotope-labeled compound)-to-tracee (endogenous compound) ratio. The glucose oxidation rates were calculated from the ^13^C_6_-glucose infusion rates, the total CO_2_ released, and the expiratory ^13^CO_2_ enrichments according to Equations (3) and (4):OR = V^13^CO_2_/Inf^13^C(3)
OR = TTR^13^CO_2_/(TTR^13^CO_2_ + 1) · *Content*CO_2_ · RMV/Inf^13^C(4)
where OR is the glucose oxidation rate, V^13^CO_2_ is the total expiratory ^13^CO_2_ release, Inf^13^C is the ^13^C infusion rate, TTR^13^CO_2_ is the expiratory tracer-to-tracee ratio of ^13^CO_2_, *Content*CO_2_ is the total expiratory CO_2_ content, and RMV is the respiratory minute volume. Inf^13^C was calculated from the ^13^C_6_-glucose i.v. infusion rate, Inf^13^C_6_-Gluc, and the number of ^13^C-labeled C atoms in the glucose molecule, N.
Inf^13^C = Inf^13^C_6_-Gluc · N = Inf^13^C_6_-Gluc · 6(5)

### 2.7. Mitochondrial Respiratory Activity

After euthanasia, heart, liver, and kidney tissues were harvested for an analysis of the mitochondrial respiratory activity via high-resolution respirometry with a Clark electrode-based system (Oxygraph 2k, OROBOROS Instruments Corp., Innsbruck, Austria), as described previously [19,27]. Briefly, immediate postmortem tissue specimens were mechanically homogenized in Mir05 (respiration medium). Mir05 is composed of 0.5 mM EGTA, 3 mM MgCl_2_·6H_2_O, 60 mM lactobionic acid, 20 mM taurine, 10 mM KH_2_PO_4_, 20 mM HEPES, 110 mM sucrose, and 1 g/l bovine serum albumin. A total of 1.5–2 mg of tissue (1.5 mg: heart, 2 mg: kidney, liver) was added to the Oxygraph chamber. Through the addition of a defined sequence of substrates and inhibitors, various states of mitochondrial function could be assessed. Complex I’s activity was determined after the addition of 10 mM pyruvate, 10 mM glutamate, 5 mM malate, and 5 mM ADP. An amount of 10 µM cytochrome c was added to check the integrity of the outer mitochondrial membrane. The maximum oxidative phosphorylation (OxPhos) was evaluated after the subsequent addition of 1 mM octanoyl–carnitine and 10 mM succinate. The leak compensation was assessed after the inhibition of the ATP synthase using 2.5 µM oligomycin, followed by the stepwise titration of the uncoupling agent carbonyl cyanide-4-(trifluoromethoxy)phenylhydrazone (FCCP, final concentration of 1.5 µM) to reach the maximum respiratory activity of the electron transfer system in the uncoupled state (ETC). Complex IV’s activity was determined after the addition of 2 mM ascorbate and 0.5 m M N,N,N′,N′-tetramethyl-*p*-phenylenediamine dihydrochloride (TMPD), in order to avoid uncontrolled autoxidation.

### 2.8. Power Analysis and Statistics

As mentioned above, the present experiments were part of a larger protocol that also included mice without co-morbidities and streptozotocin-induced diabetes type 1 [19,20]. A power analysis to detect differences anticipated a non-normal distribution of the data for the main criteria, vasopressor support and creatinine clearance. Consequently, this analysis resulted in 14 animals per study group. In the present study, three animals had to be excluded from the final data analysis. In addition, three animal experiments were not performed due to the result of an interim analysis after twenty-five animals had been analyzed. Following this interim analysis, it appeared that the vasopressor support and creatinine clearance data were unlikely to achieve significant differences. Therefore, in accordance with the “3R principle”, we stopped the experimental series to avoid the experiments yielding futile results.

The data collection was performed using Excel (Microsoft, Redmond, WA 98052, USA), while the statistical analyses were performed using Graphpad (Prism, Boston, MA, USA). The data are presented as the median and interquartile range or the mean ± standard deviation as appropriate, according to the absence/presence of a normal data distribution. The Shapiro–Wilk test was used to test for a normal distribution of data. Differences between the treatment groups were analyzed using the Mann–Whitney U rank sum test or a Student’s *t*-test as appropriate, and differences within a treatment group were analyzed using a paired Student’s *t*-test or a paired Wilcoxon rank sign test as appropriate. Differences with an alpha error below 0.05 were considered significant. A log rank Mantel–Cox test was used to determine the survival differences between the treatment groups.

## 3. Results

Three animals had to be excluded from the final data analysis due to hemothorax and/or pericardial tamponade subsequent to the blunt chest trauma, or uncontrollable bleeding during surgery. Thus, the data presented refer to the 25 mice which were analyzed (vehicle group: n = 11; Na_2_S_2_O_3_ group: n = 14). This drop-out rate of 3/28 is similar to that of previous studies conducted by our group using the same model in CSE^−/−^ mice, and was due to the severe combined trauma injury mechanism of blunt chest trauma and hemorrhagic shock as well as the extensive surgical instrumentation [20].

Figure 2 shows the Kaplan–Meier survival curve for the two experimental groups. The Na_2_S_2_O_3_-treated mice presented with a slightly higher mortality than the control animals during the observation period; however, this result did not reach statistical significance (survival time for Na_2_S_2_O_3_ group: 359.0 ± 96.9 vs. vehicle group: 379.4 ± 97.0 min; log rank test: *p* = 0.068). The hemodynamics, lung mechanics, gas exchange, acid–base status, whole-body metabolism, the kidney function at baseline, i.e., immediately after the insertion of the arterial catheter, and the kidney function at the end of the experiment are depicted in Table 1. Due to the hemorrhagic shock and the repeated intermittent blood sampling, the total hemoglobin content significantly fell over time; however, this result did not have an inter-group difference. The hemodynamics, vasopressor requirement, and acid–base balance did not differ between the two groups either. However, while no effect was observed in the vehicle group, the Na_2_S_2_O_3_ treatment was associated with a fall in PaO_2_ over time (74 ± 21 vs. 89 ± 10 mmHg; *p* = 0.089) at the end of the experiment, which coincided with a trend towards lower static thoraco-pulmonary compliance (99 (88; 114) vs. 112 (106; 123) µL/cm H_2_O; *p* = 0.071). In the vehicle group, both the peak inspiratory pressures (7 ± 1 vs. 6 ± 1 mmHg; *p* = 0.043) and the minute ventilation (875 ± 192 vs. 796 ± 152 mL/min; *p* = 0.045) required to maintain the PaCO_2_ levels within the target range could be reduced throughout the observation period. In contrast, in the Na_2_S_2_O_3_-treated mice, the minute ventilation had to be kept constant, because the whole-body CO_2_ production was higher at the end of the experiment (27 ± 4 vs. 23 ± 5 μL/g/min). However, this difference just missed statistical significance (*p* = 0.065 vs. vehicle group). The glycemia levels significantly fell over time in both groups (vehicle group: 226 ± 59 vs. 88 ± 17 mg/dL, *p* = 0.002; Na_2_S_2_O_3_ group: 203 ± 50 vs. 72 ± 17 mg/dL, *p* < 0.001), and this effect was more pronounced in the Na_2_S_2_O_3_-treated mice, which resulted in lower values in this group at the end of the experiment (72 ± 17 vs. 88 ± 17 mg/dL; *p* = 0.050). However, neither the arterial lactate levels nor the parameters of kidney function showed any significant time- or group-related effects.

The overall and individual results of the stable-isotope-based assessment of the metabolic pathways are presented in Figure 3. Leucine, urea, and glycerol, i.e., the parameters of proteolysis, lipolysis, and combined hepatic metabolic capacity and kidney function, respectively (*p* = 0.798, 0.957, and 0.111, respectively, vs. vehicle group), and the endogenous glucose production rate (*p* = 0.588 vs. vehicle group), failed to show any significant inter-group differences. In contrast, the direct, aerobic glucose oxidation (in % of the infused isotope-labeled glucose) was significantly higher in the Na_2_S_2_O_3_-treated mice (49.2 ± 9.1 vs. 56.6 ± 6.4%; *p* = 0.041 vs. vehicle group).

The overall and individual results for the maximum oxidative phosphorylation (OxPhos) as well as the maximum respiratory activity of the electron transfer system in the uncoupled state (ETC) of the immediate postmortem heart, liver, and kidney tissue specimens are shown in Figure 4. None of these parameters showed any inter-group differences.

## 4. Discussion

The present study aimed to determine whether i.v. Na_2_S_2_O_3_, as an exogenous H_2_S donor during resuscitation from combined blunt chest trauma and hemorrhagic shock in a murine model, would allow for the stabilization of hemodynamics and mitigate post-traumatic organ dysfunction in animals with pre-traumatic CS exposure by improving their energy metabolism. The main findings were that Na_2_S_2_O_3_ (i) did not exert any beneficial effect at all and (ii) was even associated with aggravated lungs, (iii) ultimately resulting in enhanced premature mortality during the pre-scheduled observation period. Among the various parameters recorded to assess the metabolic effect of Na_2_S_2_O_3_, we only observed lower glycemia levels resulting from a higher rate of direct, aerobic glucose oxidation.

Clearly, our study was not powered to detect any differences in survival, but statistical significance was only minimally missed (*p* = 0.068). Nevertheless, the present findings agree with the results seen in CSE^−/−^ mice, both without co-morbidities and with pre-existing diabetes type 1, where no beneficial outcome was observed [19,20]. These findings contrast with previous studies in mice with endotoxemia, sepsis, or cerebral ischemia, which reported improved survival in the Na_2_S_2_O_3_-treated mice [16,33,34]. However, in contrast to our present work, the animals in those studies did not receive any critical care, i.e., mechanical ventilation and/or circulatory support.

The median NoA infusion rate required to maintain the target hemodynamics was lower in the Na_2_S_2_O_3_ group; however, it clearly did not reach a statistically significant effect (*p* = 0.53). This result contrasts with the findings of our previous study on CSE^−/−^ mice treated without co-morbidities, where Na_2_S_2_O_3_ allowed the vasopressor to be significantly reduced. However, the results agree with our study in CSE^−/−^ mice with pre-existing diabetes type 1, where Na_2_S_2_O_3_ had no effect [19,20]. Taken together, these results suggest that Na_2_S_2_O_3_ allows the systemic hemodynamics to be improved only in the absence of a pre-existing co-morbidity. Interestingly, in the vehicle group, the median NoA infusion rate was 3–5 times lower than in our previous studies (median: 0.073 vs. 0.27 and 0.58 μg/kg/min) for mice without co-morbidities and with diabetes type 1, respectively [19,20]. Moreover, despite the lower NoA infusion rates, the median MAP values at the end of the experiment were lower in the vehicle-group mice in these latter studies (48 and 44 mmHg in animals without co-morbidities and with diabetes type 1, respectively) than in the present study (60 mmHg). Since all the other factors potentially influencing systemic hemodynamics were comparable in all three experimental series, it is likely that the different systemic hemodynamics were related to the pre-traumatic CS exposure. In fact, this finding agrees well with a previous study, where pre-traumatic CS exposure in CSE^−/−^ mice was associated with the highest blood pressure values when compared to a CSE deletion or CS exposure alone. The higher MAP values could be due to the increased overall sympathetic activity reported in CS-exposed mice [23].

In contrast to our aforementioned previous studies, where Na_2_S_2_O_3_ improved or had no effect on pulmonary gas exchange [19,20], it actually deteriorated PaO_2_ until the end of the experiment in the present investigation. However, this effect did not reach statistical significance. We did not measure the whole-body O_2_ consumption, but the strictly identical systemic hemodynamics suggest that the major extra-pulmonary factor influencing PaO_2_, i.e., the pulmonary arterial PO_2_, did not assume major importance. Hence, the fall in PaO_2_ was most likely due to increased intrapulmonary right-to-left shunting. This reasoning is further supported by the fact that Na_2_S_2_O_3_ administration was associated with a trend towards lower static thoraco-pulmonary compliance, suggesting more pronounced partial and/or even complete alveolar collapse. We can only speculate about any potential effect of the Na_2_S_2_O_3_ treatment on dead space ventilation; both the peak inspiratory pressures and the minute ventilation required to maintain the target PaCO_2_ levels remained unchanged throughout the observation period in the Na_2_S_2_O_3_-treated mice, while they could be reduced until the end of the experiment in the vehicle group. However, the higher CO_2_ production rate in these animals made any reduction in the invasiveness of mechanical ventilation impossible.

It is well established that, depending on its local concentration, H_2_S can either inhibit or even stimulate mitochondrial respiratory activity. In fact, during hypothermia, inhaled H_2_S was found to significantly increase the whole-body direct, aerobic glucose oxidation in mechanically ventilated mice [35]. In that experiment, inhaled H_2_S was also associated with an attenuated responsiveness of liver tissue mitochondrial respiratory activity to exogenous cytochrome c stimulation 38, indicating the protection of the outer mitochondrial membrane [36]. Finally, Na_2_S_2_O_3_ was shown to protect mitochondrial function during liver ischemia–reperfusion injury [26]. Therefore, in this study, special attention was paid to combining the analysis of metabolic pathways with tissue mitochondrial respiratory activity. While Na_2_S_2_O_3_ did not affect the endogenous glucose production rate, it was associated with a significant increase in the direct, aerobic glucose oxidation rate. In line with this, the plasma glucose levels tended to be lower and the whole-body CO_2_ production tended to be higher in the Na_2_S_2_O_3_-treated mice, but both findings just missed statistical significance (*p* = 0.066 and *p* = 0.065, respectively, vs. vehicle group). Moreover, the parameter of lipolysis, i.e., the glycerol production rate, also tended to be lower in the Na_2_S_2_O_3_ group (*p* = 0.111 vs. vehicle group). Taken together, these findings suggest that the Na_2_S_2_O_3_ treatment coincided with a metabolic switch towards the preferential use of glucose for energy metabolism rather than fatty acids. Such a switch in fuel utilization is associated with an improved energy balance [37]. We had previously observed a similar metabolic switch in mechanically ventilated hypothermic mice during H_2_S inhalation [35]. While in pigs, the oxidation of glucose rather than fatty acids enhanced the left heart contractility and improved the pressure–volume relationship [38], it is unlikely that this metabolic effect assumed importance for the heart function in our experiment; the systemic hemodynamics were strictly identical in the two groups, and the median NoA infusion rates required to achieve the target hemodynamics were not. Finally, in contrast to this potentially beneficial effect on carbohydrate utilization, Na_2_S_2_O_3_ was not associated with any modifications to the liver, kidney, or, in particular, heart tissue’s mitochondrial activity. These latter findings agree with those of our previous study on CSE^−/−^ mice without underlying co-morbidities, where Na_2_S_2_O_3_ did not influence the immediate post mortem liver and kidney mitochondrial respiration, despite the otherwise protective effect on these organs [5]. Clearly, our results are in sharp contrast to the Na_2_S_2_O_3_-related protection of liver mitochondrial function reported by others during murine liver ischemia–reperfusion injury [26]. Nevertheless, they do agree with data on rats with colon ascendens stent-induced peritonitis, where Na_2_S_2_O_3_ improved the liver microcirculation while the tissue mitochondrial function remained unaltered [10]. It should be noted that, in contrast to our present work, the animals in those studies did not receive any critical care; in particular, no circulatory support using i.v. NoA was provided, which is well established to exert major metabolic effects, both in experimental animals, e.g., mice, and in human patients [39].

### Limitations of the Study

The main limitation of our study is the fact that it suffers from the lack of a comparison between wild-type and CSE ^−/−^ mice, where different hemodynamics could have been expected. The present experiments were part of a larger protocol that also included mice without co-morbidities and streptozotocin-induced diabetes type 1 [19,20]. However, the federal authorities only approved the animal experiments in CSE^−/−^ mice, where the strongest effect of Na_2_S_2_O_3_ was expected, and, consequently, we were not allowed to include wild-type control mice in the present study as originally planned. Thus, any effect of Na_2_S_2_O_3_ in wild-type mice with pre-traumatic CS exposure remains to be assessed.

Moreover, we did not measure the blood or tissue H_2_S levels, and hence, we cannot provide evidence that Na_2_S_2_O_3_ increased the tissue H_2_S concentrations. It should be noted, however, that we studied CSE^−/−^ mice with pre-traumatic CS exposure; a genetic CSE deletion is a well-established murine model of reduced H_2_S availability [28], and pre-traumatic CS exposure per se is associated with decreased blood sulfide levels [31]. In addition, an adequate determination of H_2_S is challenging; in aqueous samples, H_2_S is present in an equilibrium between physically dissolved H_2_S and the dissolved hydrosulfide anion (HS^−^), which are considered to be the biologically active forms of the molecule. Hydrogen sulfide gas (H_2_S) can escape from water dissolution into the headspace, which makes concentration measurements even more difficult, particularly since biologically active concentrations can be as low as in the low ppb range. Furthermore, this equilibrium is pH-dependent, so that the fraction of dissolved gas vs. dissolved anions is variable. This renders concentration measurements even more complex. Consequently, no technique is available yet to directly measure the total H_2_S concentrations in biological samples. We have developed a stable, non-radioactive isotope-labeling-based gas chromatography/mass spectrometry technique to assess sulfide levels [27]. Since we have previously shown that, during porcine hemorrhage and resuscitation, a Na_2_S_2_O_3_-infusion in a substantially lower dose range resulted in three times higher sulfide blood levels than without Na_2_S_2_O_3_ administration [17], we assumed that a similar effect would be present in the murine experiment.

It could also be argued that our study does not provide data on the immune response, e.g., plasma or tissue cytokine levels, nor does it provide a histology and/or immunohistochemistry analysis. Clearly, in our previous studies, beneficial effects of Na_2_S_2_O_3_ on the immune response were observed. However, in recent studies following the same protocol in rodents without or with different co-morbidities, we were unable to confirm such a mitigation of any impaired immune reactions. Therefore, we did not expect there to be any major effects of Na_2_S_2_O_3_ on mice with pre-existing COPD after trauma.

Lastly, our study period was limited to 8 h. However, once patients survive initial resuscitation and surgery, they usually undergo critical care for several days, if not weeks. Consequently, our study can only be discussed for the early phase of resuscitation, but not for the phase of prolonged critical care. Together with the above-mentioned lack of data on Na_2_S_2_O_3_ applications in wild-type mice, the clinical use of Na_2_S_2_O_3_ as an adjunct to resuscitation remains an ongoing issue.

## 5. Conclusions

In this study on CSE^−^/^−^ mice with pre-existing CS exposure, exogenous H_2_S supplementation using Na_2_S_2_O_3_ during resuscitation from combined blunt chest trauma and hemorrhage was associated with increased direct, aerobic glucose oxidation, suggesting a switch in energy metabolism towards preferential carbohydrate utilization. However, the lung mechanics, gas exchange, and ultimately, survival tended to be deteriorated.

## Figures and Tables

**Figure 1 biomedicines-12-02581-f001:**
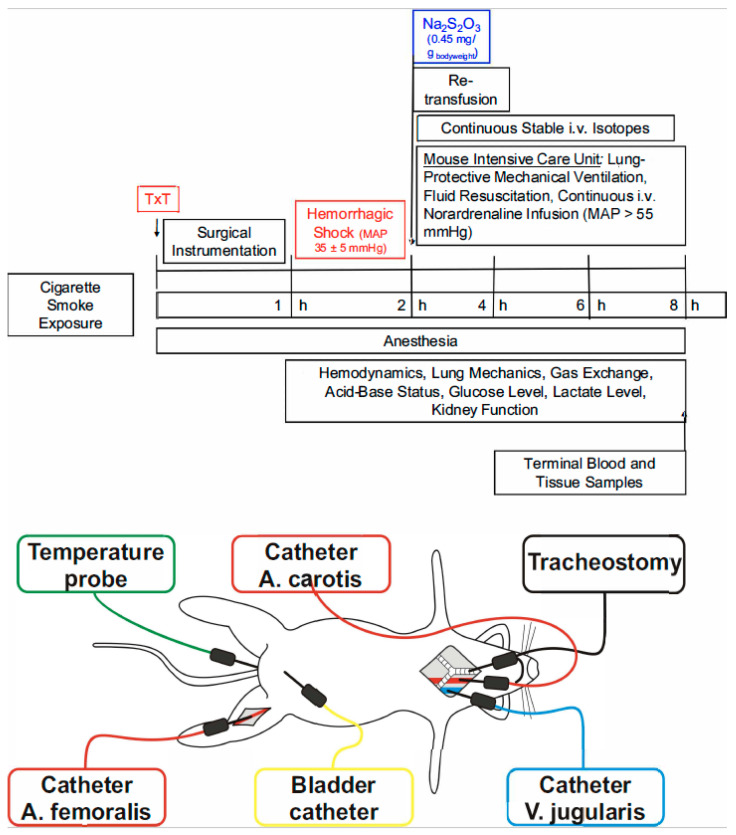
Experimental setup, timeline, surgical instrumentation, and experimental protocol. MAP, mean arterial pressure; Na_2_S_2_O_3_, sodium thiosulfate; TxT, chest trauma; i.v., intravenous.

**Figure 2 biomedicines-12-02581-f002:**
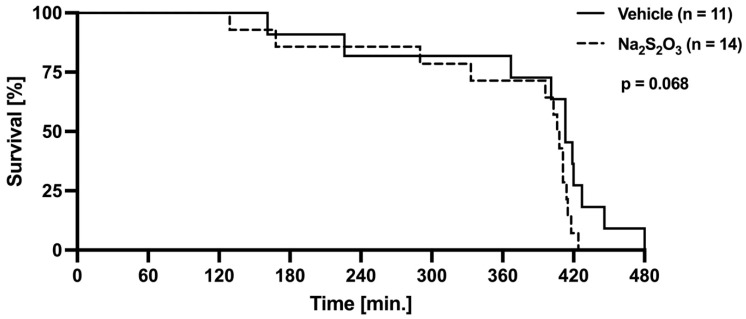
Kaplan–Meier survival curve in the vehicle group (solid line) and Na_2_S_2_O_3_ group (dotted line). Time “0” refers to the start of the observation period, with the initiation of blunt chest trauma. No significant differences were observed (log rank Mantel–Cox test, *p* = 0.068).

**Figure 3 biomedicines-12-02581-f003:**
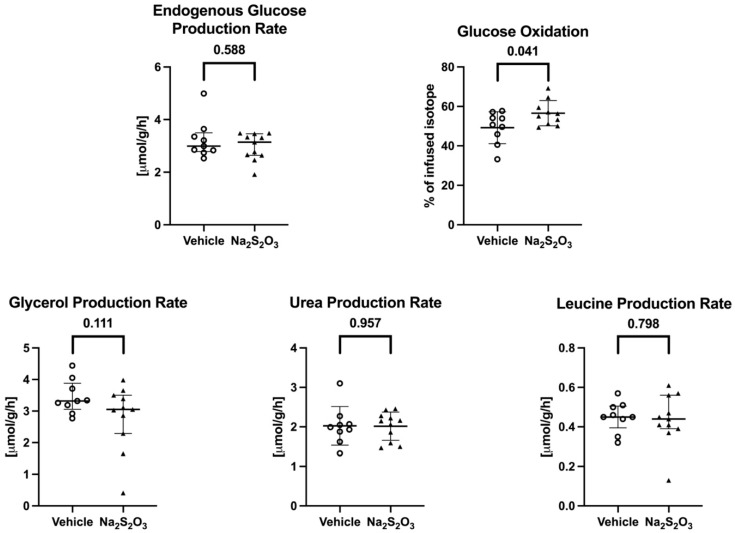
The evaluation of metabolic pathways. The individual results as well as the median and interquartile range (endogenous glucose production rate) or mean ± standard deviation (glucose oxidation; glycerol, urea, and leucine production rates) are shown, according to the absence/presence of a normal data distribution for the metabolic parameters as assessed using stable, non-radioactive isotope-labeled substrates (glucose, glycerol, leucine, and urea) between mice in the vehicle (open circles) and the Na_2_S_2_O_3_ (solid triangles) groups. Differences between the two treatment groups were analyzed using the Mann–Whitney U rank sum test or a Student’s *t*-test as appropriate.

**Figure 4 biomedicines-12-02581-f004:**
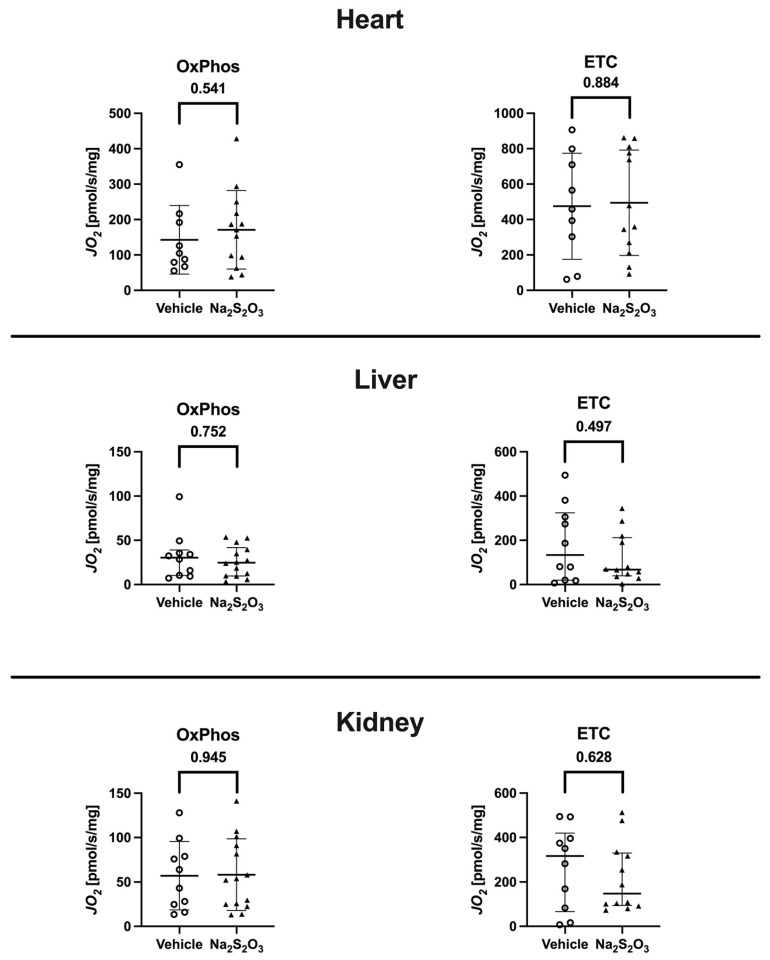
The evaluation of mitochondrial activity. The individual results as well as the median and interquartile range (liver OxPhos and ETC, kidney ETC) or mean ± standard deviation (heart OxPhos and ETC, kidney OxPhos) are shown according to the absence/presence of a normal data distribution for the parameters of mitochondrial respiratory activity in immediate postmortem specimens of the heart, liver, and kidney. The oxidative phosphorylation (OxPhos) (left panel) and maximal electron transfer capacity in the uncoupled state (ETC, right panel) are presented for mice in the vehicle (open circles) and Na_2_S_2_O_3_ (solid triangles) groups. Differences between the two treatment groups were analyzed using the Mann–Whitney U rank sum test or a Student’s t-test as appropriate.

**Table 1 biomedicines-12-02581-t001:** Parameters of hemodynamics, lung mechanics, gas exchange, acid–base status, whole-body metabolism, and kidney function. All the data are shown as the median (interquartile range) or mean ± standard deviation as appropriate, according to the absence/presence of a normal data distribution; § *p* < 0.05 vs. baseline. Differences between the two treatment groups were analyzed using the Mann–Whitney U rank sum test or a Student’s *t*-test as appropriate; differences within a treatment group were analyzed using a paired Student’s *t*-test or a paired Wilcoxon rank sign test as appropriate.

Parameter	Treatment	Baseline	End of Experiment
Heart rate [1/min]	Vehicle	455 ± 59	446 ± 45
	Na_2_S_2_O_3_	458 ± 68	454 ± 103
Mean arterial pressure [mm Hg]	Vehicle	70 (61; 81)	60 (55; 71)
	Na_2_S_2_O_3_	71 (61; 86)	60 (57; 70)
Noradrenaline infusion rate [μg/kg/min]	Vehicle	-	0.073 (0; 1.236)
	Na_2_S_2_O_3_	-	0.039 (0; 0.161)
Hemoglobin	Vehicle	8.4 ± 1.0	6.7 ± 0.8 §
[g/dL]	Na_2_S_2_O_3_	8.6 ± 1.6	6.6 ± 1.2 §
Minute ventilation [mL/min]	Vehicle	875 ± 192	796 ± 152 §
	Na_2_S_2_O_3_	921 ± 133	848 ± 212
Positive end-expiratory pressure [cmH_2_O]	Vehicle	3 (3; 3)	3 (3; 3)
	Na_2_S_2_O_3_	3 (3; 3)	3 (3; 3)
Peak inspiratory pressure [cmH_2_O]	Vehicle	7 ± 1	6 ± 1 §
	Na_2_S_2_O_3_	7 ± 2	7 ± 1
Thoraco-pulmonary	Vehicle	106 (93; 112)	112 (106; 123)
compliance [µL/cm H_2_O]	Na_2_S_2_O_3_	98 (93; 108)	99 (88; 114) *p* = 0.071 vs. Vehicle
Arterial PO_2_ [mm Hg]	Vehicle	86 ± 16	89 ± 10
	Na_2_S_2_O_3_	89 ± 14	74 ± 21 § *p* = 0.089 vs. Vehicle
Arterial PCO_2_ [mm Hg]	Vehicle	35 ± 9	35 ± 6
	Na_2_S_2_O_3_	33 ± 8	40 ± 13
Arterial pH	Vehicle	7.25 (7.21; 7.36)	7.33 (7.29; 7.37)
	Na_2_S_2_O_3_	7.30 (7.24; 7.36)	7.29 (7.25; 7.34)
Arterial base excess [mmol/L]	Vehicle	−9.4 ± 3.7	−7.1 ± 3.3
	Na_2_S_2_O_3_	−9.9 ± 4.3	−8.9 ± 4.7
CO_2_ production	Vehicle	n.d.	23 ± 5
[μL/g/min]	Na_2_S_2_O_3_	n.d.	27 ± 4 *p* = 0.065 vs. Vehicle
Arterial glucose [mg/dL]	Vehicle	226 ± 59	88 ± 17 §
	Na_2_S_2_O_3_	203 ± 50	72 ± 17 § *p* = 0.050 vs. Vehicle
Arterial lactate [mmol/L]	Vehicle	1.2 (0.9; 1.7)	0.9 (0.8; 1.3)
	Na_2_S_2_O_3_	1.1 (1.1; 1.3)	1.1 (0.7; 2.8)
Plasma creatinine	Vehicle	n.d.	1.2 (0.9; 1.4)
[μg/mL]	Na_2_S_2_O_3_	n.d.	1.3 (1.0; 1.5)
Plasma urea	Vehicle	n.d.	38 (27; 45)
[mg/dL]	Na_2_S_2_O_3_	n.d.	34 (28; 45)
Creatinine clearance	Vehicle	n.d.	0.5 (0.4; 0.7)
[mL/min]	Na_2_S_2_O_3_	n.d.	0.6 (0.5; 1.0)

## Data Availability

The datasets used during the current study are available from the corresponding authors upon reasonable request.

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
