# Peer review of "Metabolic Effects of Sodium Thiosulfate During Resuscitation from Trauma and Hemorrhage in Cigarette-Smoke-Exposed Cystathionine-γ-Lyase Knockout Mice"

_biomedicines, 2024, doi:10.3390/biomedicines12112581_

Round 1

Reviewer 1 Report

Comments and Suggestions for Authors

The manuscript deals with the topic of high interest. It is focused on the evaluation of the metabolic effects of sodium thiosulfate as a source of H2S during resuscitation from blunt chest trauma and hemorrhagic shock in cigarette smoke-exposed cysta-thionine-γ-lyase knockout mice. The  body metabolic pathways and mitochondrial respiratory activity has been studied in details. The investigations carried out have shown that Na2S2O3 promotes increased direct aerobic oxidation of glucose that can indicate realization of carbohydrate utilization during the energy metabolism. At the same time, the Na2S2O3 treatment makes worse lung mechanics and gas exchange, and leads to shorter survival.

The manuscript is very well prepared. The experiment is logically designed and described in details. The results and discussion are meaningful. Good comparison to other studies on the topic is performed. The critical remarks presented in the subsection "Limitations of the study" are clearly show further problem to be solved. Conclusions are fully supported with the data obtained.

Thus, the manuscript is of high scientific quality and can be accepted to publication after minor revision. The technical points to be corrected are presented below.

1. Abstract, the font size to be unified.

2. The references in the text to be divided from the text where they appear with the space. Corrections are required throughout the manuscript.

3. Line 91, the reference [33] suddenly appeared after 28. Please, clarify this point and check the reference numbering.

4. Lines 212-213, ...tert-... to be in italics.

5. Line 258, N,N,N',N'-Tetramethyl-p-phenylenediamine dihydrochloride replace with N,N,N',N'-tetramethyl-p-phenylenediamine dihydrochloride.

6. All abbreviations used 1-2 times throughout the manuscript to be removed in order to improve readability of the text.

7. Fig. 2, X axis title, remove dot after min which is a standard designation for minute.

Author Response

Dear Reviewer,

thank you for your time and effort to review and improve our manuscript. We have addressed all suggestions made (see below).

Sincerely yours,

The Authors

The manuscript deals with the topic of high interest. It is focused on the evaluation of the metabolic effects of sodium thiosulfate as a source of H2S during resuscitation from blunt chest trauma and hemorrhagic shock in cigarette smoke-exposed cysta-thionine-γ-lyase knockout mice. The body metabolic pathways and mitochondrial respiratory activity has been studied in details. The investigations carried out have shown that Na2S2O3 promotes increased direct aerobic oxidation of glucose that can indicate realization of carbohydrate utilization during the energy metabolism. At the same time, the Na2S2O3 treatment makes worse lung mechanics and gas exchange, and leads to shorter survival.

The manuscript is very well prepared. The experiment is logically designed and described in details. The results and discussion are meaningful. Good comparison to other studies on the topic is performed. The critical remarks presented in the subsection "Limitations of the study" are clearly show further problem to be solved. Conclusions are fully supported with the data obtained.

Thus, the manuscript is of high scientific quality and can be accepted to publication after minor revision. The technical points to be corrected are presented below.

  1. Abstract, the font size to be unified.

Font Size within the Abstract was unified.

  1. The references in the text to be divided from the text where they appear with the space. Corrections are required throughout the manuscript.

We have inserted a space between the text and references as appropriate.

  1. Line 91, the reference [33] suddenly appeared after 28. Please, clarify this point and check the reference numbering.

Thank you for making us aware of this issue. We have updated the reference numbering.

  1. Lines 212-213, ...tert-... to be in italics.

Adjusted accordingly.

  1. Line 258, N,N,N',N'-Tetramethyl-p-phenylenediamine dihydrochloride replace with N,N,N',N'-tetramethyl-p-phenylenediamine dihydrochloride.

Adjusted accordingly.

  1. All abbreviations used 1-2 times throughout the manuscript to be removed in order to improve readability of the text.

Thank you for this suggestion. We have adjusted the use of abbreviations and hope, the manuscript’s readability is improved.

  1. 2, X axis title, remove dot after min which is a standard designation for minute.

Figure 2 was adjusted accordingly.

Reviewer 2 Report

Comments and Suggestions for Authors

The manuscript currently under review delves into the metabolic repercussions of sodium thiosulfate (Na2S2O3), an exogenous hydrogen sulfide donor, during the resuscitation phase from trauma and hemorrhagic shock in cystathionine-γ-lyase (CSE) knockout mice previously exposed to cigarette smoke. The study intriguingly reveals that Na2S2O3-treated mice, in comparison to the control cohort, demonstrated an elevated rate of direct aerobic glucose oxidation. However, these mice also presented with diminished lung function and reduced survival times. The authors have innovatively harnessed the CSE knockout mouse model, providing a distinctive lens through which to examine the role of exogenous H2S in the context of trauma and hemorrhage. The findings of this research contribute to a more nuanced comprehension of the metabolic shifts prompted by Na2S2O3. Ultimately, the study adeptly tackles the implications of pre-traumatic cigarette smoke exposure. The assertions and aims of the study are in harmony with the scope of Biomedicines. This manuscript is considered suitable for publication after minor revisions.

Comments:

1.     In section 2.2, could the authors clarify whether a control group with wild-type mice was utilized to evaluate the effects of Na2S2O3 within a normal metabolic context?

2.     Regarding the assessment of Na2S2O3's impact on H2S concentrations, did the authors quantify H2S levels in both blood and tissues to substantiate their findings?

3.     The section 3 (results) is too short but the section 4 (discussion) is too long. To enhance the clarity and balance of the manuscript, the authors might consider revising sections 3 and 4 together to ensure a proportionate and coherent flow of information.

4.     A meticulous proofreading of the entire manuscript is advised to rectify any grammatical inaccuracies or typographical errors, with special diligence in the reference section to maintain academic precision and uniformity.

Author Response

Dear Reviewer,

thank you for your time and effort to review and improve our manuscript. We have addressed all suggestions made (see below).

Sincerely yours,

The Authors

The manuscript currently under review delves into the metabolic repercussions of sodium thiosulfate (Na2S2O3), an exogenous hydrogen sulfide donor, during the resuscitation phase from trauma and hemorrhagic shock in cystathionine-γ-lyase (CSE) knockout mice previously exposed to cigarette smoke. The study intriguingly reveals that Na2S2O3-treated mice, in comparison to the control cohort, demonstrated an elevated rate of direct aerobic glucose oxidation. However, these mice also presented with diminished lung function and reduced survival times. The authors have innovatively harnessed the CSE knockout mouse model, providing a distinctive lens through which to examine the role of exogenous H2S in the context of trauma and hemorrhage. The findings of this research contribute to a more nuanced comprehension of the metabolic shifts prompted by Na2S2O3. Ultimately, the study adeptly tackles the implications of pre-traumatic cigarette smoke exposure. The assertions and aims of the study are in harmony with the scope of Biomedicines. This manuscript is considered suitable for publication after minor revisions.

Comments:

  1. In section 2.2, could the authors clarify whether a control group with wild-type mice was utilized to evaluate the effects of Na2S2O3within a normal metabolic context?

Unfortunately, the federal authorities did not allow a wildtype control group to explore the effects of Na2S2O3 in an unaltered organism. A statement about the absence of a wildtype control group is mentioned in section 2.2 lines 119-123 as well as in section 4 lines 449-456.

  1. Regarding the assessment of Na2S2O3's impact on H2S concentrations, did the authors quantify H2S levels in both blood and tissues to substantiate their findings?

In biological, aqueous samples H2S is present in an equilibrium between the physically dissolved gas dissolved hydrosulfide anion (HS-), which are considered the biologically active forms of the molecule. The gaseous form of hydrogen sulfide, the hydrogen sulfide gas (H2S), can escape from water dissolution into the headspace, which makes concentration measurements extremely difficult, in particular, since biologically active concentrations are as low as in the low ppb-range (for details see: McCook O, Radermacher P, Volani C, Asfar P, Ignatius A, Kemmler J, Möller P, Szabó C, Whiteman M, Wood ME, Wang R, Georgieff M, Wachter U. H2S during circulatory shock: Some unresolved questions. Nitric Oxide Biol Chem 2014;41:48-61 and Radermacher P, Calzia E, McCook O, Wachter U, Szabo C. To the Editor. Shock 2021;55:138–9). Moreover, the above-mentioned equilibrium is pH-dependent, so that the fraction of dissolved gas vs. dissolved anion is variable, which renders concentrations measurements even more complex. Consequently, so far, no technique is available to directly measure total H2S concentrations in biological samples. Nevertheless, we have developed a stable, non-radioactive isotope labelling-based gas chromatography/mass spectrometry-technique to assess sulfide levels (for example see Hartmann C, Gröger M, Noirhomme JP, Scheuerle A, Möller P, Wachter U, Huber-Lang M, Nussbaum B, Jung B, Merz T, McCook O, Kress S, Stahl B, Calzia E, Georgieff M, Radermacher P, Wepler M. In depth-characterization of the effects of cigarette smoke exposure on the acute trauma response and hemorrhage in mice. Shock 2019;51:68-77). Since we had previously shown that Na2S2O3-infusion in substantially lower dose range had resulted in three times higher sulfide blood levels than without Na2S2O3 administration (see Datzmann T, Hoffmann A, McCook O, Merz T, Wachter U, Preuss J, Vettorazzi S, Calzia E, Gröger M, Kohn F, Schmid A, Radermacher P, Wepler M. Effects of sodium thiosulfate (Na2S2O3) during resuscitation from hemorrhagic shock in swine with preexisting atherosclerosis. Pharmacol Res 2020;151:104536), we assumed that a similar effect would be present in the murine experiment. Therefore, we did not measure sulfide levels in this experiment.

  1. The section 3 (results) is too short but the section 4 (discussion) is too long. To enhance the clarity and balance of the manuscript, the authors might consider revising sections 3 and 4 together to ensure a proportionate and coherent flow of information.

Thank you for this comment. We expanded the results section and in order to provide a more balanced structure of our manuscript. However, we only enlarged this section slightly to avoid unnecessary redundancies to data already mentioned in tables or figures. Although we tried to streamline the discussion section, we felt the need to add a statement about the evaluation of H2S concentrations to the limitations. We tried our best to optimize our manuscripts structure and hope, you will find the adjustments sufficient.

  1. A meticulous proofreading of the entire manuscript is advised to rectify any grammatical inaccuracies or typographical errors, with special diligence in the reference section to maintain academic precision and uniformity.

We double-checked the entire manuscript including the reference section for typographical and grammatical errors. We hope, we have addressed any issues and apologize for any potential remaining mistakes.